# Identification of Ground Intrusion in Underground Structures Based on Distributed Structural Vibration Detected by Ultra-Weak FBG Sensing Technology

**DOI:** 10.3390/s19092160

**Published:** 2019-05-09

**Authors:** Weibing Gan, Sheng Li, Zhengying Li, Lizhi Sun

**Affiliations:** 1National Engineering Laboratory for Fiber Optic Sensing Technology, Wuhan University of Technology, Wuhan 430070, Hubei, China; ganweibing@whut.edu.cn; 2School of Information Engineering, Wuhan University of Technology, Wuhan 430070, Hubei, China; zhyli@whut.edu.cn; 3Department of Civil and Environmental Engineering, University of California, Irvine, CA 92697-2175, USA; lsun@uci.edu

**Keywords:** subway tunnel safety, ground intrusion detection, ultra-weak FBG, distributed vibration, dynamic measurement

## Abstract

It is challenging for engineers to timely identify illegal ground intrusions in underground systems such as subways. In order to prevent the catastrophic collapse of subway tunnels from intrusion events, this paper investigated the capability of detecting the ground intrusion of underground structures based on dynamic measurement of distributed fiber optic sensing. For an actual subway tunnel monitored by the ultra-weak fiber optic Bragg grating (FBG) sensing fiber with a spatial resolution of five meters, a simulated experiment of the ground intrusion along the selected path was designed and implemented, in which a hydraulic excavator was chosen to exert intrusion perturbations with different strengths and modes at five selected intrusion sites. For each intrusion place, the distributed vibration responses of sensing fibers mounted on the tunnel wall and the track bed were detected to identify the occurrence and characteristics of the intrusion event simulated by the discrete and continuous pulses of the excavator under two loading postures. By checking the on-site records of critical moments in the intrusion process, the proposed detection approach based on distributed structural vibration responses for the ground intrusion can detect the occurrence of intrusion events, locate the intrusion ground area, and distinguish intrusion strength and typical perturbation modes.

## 1. Introduction

As an important carrier of the urban population, the subway system has greatly eased the pressure of ground transportation. In recent decades, research on early warning and treatment of various hazards that may affect the safety of subways has attracted widespread attention. Due to fewer indicators and concise detection principles, compared with structural safety monitoring of subway infrastructure, more commercial applications have emerged in the field of subway fire monitoring. Overviews and applications related in this area were reported in [1,2,3]. However, when long and large range needs to be considered, especially for subway tunnels, it is still a great challenge to find viable measures to meet the diverse needs of structural safety monitoring.

Since the tunnel lining structure generally uses concrete as the construction material, cracks on the lining surface are often used to reflect the safety status of a subway tunnel. One method based on digital images to detect cracks was reported by Zhang et al. [4]. For the indicators of temperature and strain, Ye et al. [5] described the study of tunnel safety monitoring during the construction stage based on quasi-distributed sensing technology of a limited number of fiber Bragg grating (FBG) sensors. Recently, references [6,7] reported some research advances in FBG-based sensors that combine the Internet of things or 3D printing techniques to detect damage or movement of underground structures. In addition to the conventional indicator, the study conducted in [8] indicates that some influencing factors, such as buried depth and operation age should also be collected when assessing the state of the tunnel. Reference [9] pointed out that particular emphasis should be paid on time and space-continuous monitoring for environmental and geotechnical underground structures. However, the above studies were primarily based on discrete response information with respect to time or space, which is difficult to meet the real-time requirements of structural safety monitoring for the entire line of the actual underground structure, such as a subway. Due to the advantages of large-scale monitoring, high sensitivity, and multiplexing capacity, distributed fiberoptic sensing technology is widely considered to be an ideal means for the safety monitoring of tunnel structures. Dewynter et al. [10] indicated the feasibility of continuous monitoring of soil movements while tunneling based on Brillouin optical time domain reflectometry (BOTDR) technology. Recent studies on using BOTDR to trace the distributed strain to secure tunnel safety can be found in [11,12,13,14].

Although distributed sensing technology provides a viable way for understanding the responses of tunnel structures in view of continuous time and space, existing studies primarily focus on the static measurement based on strain or temperature. In the past decades, research of distributed dynamic measurement [15] was mainly based on distributed acoustic sensing (DAS) techniques [16,17], which have been another research hotspot in the field of engineering monitoring. In the railway fields, DAS techniques were researched for railway perimeter security [18] and condition monitoring of the train and rail [19]. In geophysical engineering, the need and application concerning simultaneous vibration and temperature sensing technology based on DAS were reviewed in [20]. Moreover, Rao [21] reported the feasibility of using DAS technology to monitor the illegal or unauthorized third-party intrusion (TPI) in oil pipelines. However few studies pay attention to the impact of ground construction on the safety of underground structures covering a long-distance range through distributed dynamic measurements. He et al. [22] proposed that by processing images taken by unmanned aerial vehicle was a feasible way to detect ground drilling construction which may affect tunnel safety. However, this method is undoubtedly susceptible to climatic conditions and occlusion of ground buildings. Moreover, the method based on the analysis of UAV-images is still difficult to meet the timely warning needs of the entire subway line. Compared with oil pipelines, subway tunnels have deeper buried depths, and more complicated boundary conditions and load propagation paths. Therefore, different test accuracy, sensitivity, response speed, signal-to-noise ratio (SNR), and other parameters need to be considered when using DAS technology to tackle the similar need for these two different fields. This may be the reason for less reports on DAS-based tunnel TPI.

Comparing with DAS technology using ordinary optic fiber, ultra-weak FBG array based on the draw tower [23,24] using sensing optic fiber, integrates both advantages of fiber optic point sensors and distributed sensors. This technology is an alternative way to achieve high-precision, fast, and wide coverage distributed measurement. Previous research around this technology focused more on monitoring strain, temperature or strain-based deformation for the object of interest [25,26]. In addition, a multi-parameter measurement system based on ultra-weak FBG array with sensitive material was proposed in [27]. However, all the research is still limited to static indicators. Actually, ultra-weak FBG array is also adept to perform dynamic monitoring [28] in addition to the above positive characteristics usually witnessed in static measurement. From the reports in [15,29,30,31,32], the comparison results in Table 1 reveal that the ultra-weak FBG array can be not only used for both static and dynamic measurements, but also has higher SNR than that of DAS sensors. Moreover, higher SNR often leads to better sensing performances, such as higher measurement accuracy, faster response time, and simpler detection circuit, so ultra-weak FBG array is more suitable than DAS when dealing with distributed vibration and other scenarios requiring high-speed measurement. Therefore, based on such performance advantages in distributed dynamic measurement, this paper explores the detection capability of ultra-weak FBG sensing array fabrication by the draw tower for ground intrusion. The experimental results of detecting the ground intrusion event of an actual subway tunnel were reported. The detection approach of ground intrusion is the second part of this paper, followed by the details on the design and implementation of a field experiment. Finally, the effectiveness of the proposed method to detect and identify a simulated intrusion is discussed based on the on-spot experimental results from the ultra-weak FBG distributed sensing technology.

## 2. Detection Methodology of Ground Intrusion

Figure 1 illustrates the distributed vibration sensing principle used to detect ground intrusion. The phenomenon of light interference caused by the reflection signals of two adjacent ultra-weak FBGs is used to detect the vibration of the object of interest. Here, the ultra-weak FBG is regarded as a mirror, and *L* represents the distance that causes light interference. The spatial resolution of the distributed vibration along the sensing fiber is typically determined by the parameter *L*. The sensitivity and the frequency response of the vibration signal measured by the strain-induced phase variation between two ultra-weak FBGs are improved by the interferometer. Here, Faraday rotating mirrors are utilized in the demodulation process of ultra-weak FBG array to suppress the polarization effect. Moreover, the 3-by-3 coupler phase demodulation algorithm is used to reconstruct the time domain signal, and restore the phase information of the vibration signal, through which the interrogation of the vibration frequency and amplitude can be realized. Further, optical time domain reflectometry technique is utilized to achieve vibration localization.

The high sensitivity of large-scale ultra-weak FBGs and the corresponding demodulation system of high speed [33] make the sensing fiber particularly suitable for locating abnormal perturbations occurring within a long-distance range. In addition, the previous study [34] revealed the repeatability of such a sensor is around 3.41 nε. When an illegal ground intrusion event occurs, the propagation path of the intrusion load from the ground to the tunnel wall and track bed can generally be demonstrated, as shown in Figure 2. Based on this assumption, the study used armored distributed sensing fibers to measure the distributed vibration of the tunnel wall and the track bed. Five-meter equidistance between adjoining FBGs along the sensing fiber determined the spatial resolution of the detection target, and this resolution almost can meet the positioning accuracy requirement for an actual subway tunnel. The approach used to quickly indicate whether a ground intrusion occurs was conducted by monitoring the distributed structural vibration responses along the tunnel and analyzing the difference in responses between the immediate state and the normal baseline state. Since the light interference region indicated by the address of ultra-weak FBG can be interrogated with the time- and wavelength-division multiplexing method [35,36] and has a corresponding relationship with the mileage information of the monitoring structure, locating the intrusion can be achieved by identifying the light interference region corresponding to the abnormal vibration responses.

## 3. Experimental Design and Implementation

### 3.1. Engineering Background of the Experimental Scheme

An actual tunnel structure (Wuhan Metro Line 7) was used in this study. Before the operation of the subway, the ultra-weak FBG sensing fibers were installed on the structural surfaces of the tunnel wall and the track bed. It covered a range of nearly three kilometers, aiming to detect the distributed structural vibration response of the monitoring zones. Figure 3 displays the actual layout of sensing fibers on the spot. The real-time vibration responses with 1 kHz sampling rate were fully transmitted back to the platform monitoring center and processed by the demodulator and servers. According to the spatial resolution of the sensing fiber and the on-spot layout of the tunnel structure, more than 500 vibration regions along the tunnel wall and the track bed can be distinguished based on the interrogated address of the light interference.

### 3.2. Design of the Ground Intrusion

The intrusion perturbation was simulated by drilling ground through a small hydraulic excavator. According to the on-spot survey and structural design blueprints of the subway, both the available ground region for the simulated perturbation and the facade relationship between the ground and underground tunnel structure were taken into account to determine the intrusion sites and path, as illustrated in Figure 4. The plane distances between the underground tunnel and five intrusion sites correspond to the right part of the plot provided in Figure 4. Here, based on design and survey data, the average buried depth of the tunnel under the selected area was approximately 22.6 m. From the facade, the position P1 was placed just above the sensing fiber that monitored the tunnel wall.

### 3.3. Implementation of the Intrusion Perturbation

Two types of perturbation postures of the experiment excavator displayed in Figure 5 were set to simulate the different strengths of the ground intrusion. Discrete and continuous pulses were applied sequentially for each perturbation posture to simulate different intrusion modes. In order to reduce the damage of the pavement, when perturbations were applied, a steel plate was placed in advance at each position given in Figure 4.

All tests were scheduled to be carried out in the early hours to reduce interference with ground traffic. As depicted in Figure 4, field tests were sequentially performed at the positions of P1, P2, P3, P4, and P5. The purpose of considering varied distance was to explore the identifiability of distributed vibration responses under different strengths and modes. After the excavator reached the designated invasion site, discrete and continuous pulses were applied sequentially in the two postures shown in Figure 5. Perturbation in each place lasted for 1 to 1.5 min. In addition to the duration time for each specified intrusion point, other critical moments in each process were also recorded, including moments about excavator movement, loading posture adjustment, and pre-intrusion repositioning.

## 4. Result Analysis and Discussion

This section reports the characteristics of distributed vibration responses of underground tunnel structures under simulated intrusions. The ability to detect and distinguish the strength and mode of intrusion loads with different distances was investigated and discussed.

### 4.1. Responses of the Whole Intrusion Process

The #159 vibration zone of the tunnel structure just below the invasion site P1 was taken as an example to illustrate the detectability of the dynamic structure responses to the intrusion load. The vibration responses of the tunnel wall and the track bed under the perturbations of multiple intrusion sites are shown in Figure 6 and Figure 7, in which dotted lines based on the field records mark the whole process of each intrusion site. It is shown that with the increase of the distance between the perturbation sites and the tunnel, the identification effect of intrusion based on the amplitude feature of structural vibration responses gradually decreased. Specifically, the results at position P5 showed that it became difficult to distinguish the vibration responses caused by the intrusion load from those of the excavator movement and pre-intrusion repositioning. Further, the response magnitude of the track bed in each loading process was significantly smaller than that of the tunnel wall. This phenomenon was consistent with the assumption described in Figure 2, indicating the dissipation of intrusion loads during propagation from the tunnel wall to the track bed. In addition, this indicated that the response of the tunnel wall was more suitable for inferring ground intrusion.

### 4.2. Identifiability of Intrusion Characteristics

The above study discussed the relationship between the vibration response of#159 zone regarded as the most conducive to receiving perturbations and ground intrusions at different distances. By further focusing on a complete intrusion process of one intrusion site in Figure 6, the feasibility of identifying the characteristics of the simulated intrusion load was discussed. Figure 8 depicts the details of the intrusion process of the position P1 marked in Figure 6, which clearly distinguishes the detailed features of each stage of the simulated intrusion loading. During the test, the steel plate used for protecting the pavement in each intrusion site was only temporarily placed rather than fixed. This resulted in the phenomenon that the plate often bounced off and deviated from the initial position, especially in the first perturbation stage of the discrete pulse. In order to address this problem that occurred during pulse loading, the drill bit was typically used to reposition the plate, so that it was apparent from the first perturbation phase of Figure 8 that the amplitude uniformity of the vibration response was inconsistent with theoretical expectations. For the second perturbation stage, the duration of the discrete pulses was short because of the difficulty in maintaining the posture stability of the excavator. Due to this reason, there was no obvious increment in the magnitude of the response in this stage. The increase in load strength due to the change in the perturbation posture was apparent in the continuous pulse process of the second perturbation posture shown in Figure 8, although the response amplitude of the first half of the continuous pulse still deviated from the expectation. The deviation discrepancy was mainly attributed to disturbances caused by the unstable posture of the excavator and the process of re-adjusting the plate.

Moreover, both the excavator movement and the repositioning of the drill bit during the adjustment process can be observed in Figure 8, which was more obvious in the time–frequency spectrum shown in Figure 9, based on short-time Fourier transform. As can be seen from Figure 9a, different stages of the simulated intrusion corresponded to different frequency response ranges of the tunnel wall. For the excavator movement, the structure frequency was approximately 8 Hz. When the pulse loads acted on the ground, more structure frequencies were presented. Also, Figure 9a indicates the fact that the structure frequencies caused by each pulse mode were identical, in which the discrete pulses stimulated frequencies around 37 Hz, 47 Hz, and 115 Hz. Instead, continuous pulses made more frequencies with a maximum frequency around 267 Hz. Furthermore, as indicated in Figure 9b, the vibration energy caused by discrete or continuous pulses in the second perturbation posture was apparently greater than those in the first perturbation posture. These experimental results were in agreement with the expectations of the design and further demonstrated the ability of the proposed method of identifying typical load patterns.

### 4.3. Detection Range of the Simulated Intrusion

The coordinate system depicted in Figure 10 was defined to analyze the influence range of the simulated intrusion. The *XOY* and *XOY’* represent the planes of the ground and the underground buried tunnel, respectively, where *X’*, *Y*, and *Z* axes indicate the tunnel mileage, intrusion path, and depth from the ground, respectively. The origins of the planes *XOY* and *X’O’Y’* were set to the intrusion position P1 and the tunnel #159 area, respectively.

Figure 11, Figure 12, Figure 13, Figure 14 and Figure 15 plot the waterfall diagrams of the distributed structural vibration responses of 13 consecutive monitoring zones of the tunnel wall with a length of 65 m along the *X’* axis, where the response of the #159 area in each waterfall diagram was set in the center and highlighted in a different color. Comparisons among Figure 11 show that at least 9 test zones with the #159 zone as the symmetry center had the ability to identify the simulated intrusion. Since the position P5 was farther away from the tunnel than other intrusion sites, it was not always easy to distinguish the simulated intrusion based on weak responses for most of the monitoring zones shown in Figure 15. Figure 11, Figure 12, Figure 13, Figure 14 and Figure 15 also revealed that the amplitude of the distributed vibration response along the defined *X’* axis and the response symmetry of the monitoring zones centered on the #159 zone gradually became weak as the intrusion site moved away from the tunnel. Here, the asymmetry may be related to the fact reflected in Figure 4 that the designed intrusion path was not completely orthogonal to the tunnel.

To further analyze the characteristics of the distributed vibration responses along the *Y* and *X’* axes, the intrusion response sensitivity *k* was defined as,
(1)$$k=\frac{r_{\mathrm{intrusion}}}{r_{\mathrm{stationary}}}$$
where $r_{\mathrm{intrusion}}$ and $r_{\mathrm{stationary}}$ represented the root mean square values of the vibration signals of the intrusion state and the stationary state, respectively. The duration used to determine $r_{\mathrm{intrusion}}$ and $r_{\mathrm{stationary}}$ came from the field records. In order to search for potential regularity, we computed the *k* values of 13 consecutive regions in five simulated intrusion tests and obtained a *k* matrix with a shape of 5 by 13. Then, the medians of *k* were calculated based on the *k* matrix according to the intrusion site and the tunnel monitoring zone, respectively. After this statistical operation, the median-based distributions along the defined *Y* and *X’* axes are shown in Figure 16.

As shown in Figure 16a, the medians of *k* under the perturbation at each site except P5 were greater than 2.5. It can also be observed from the figure that the bar peak occurred at P2 rather than P1. One possibility for this unforeseen situation was the different ground conditions at the simulated intrusion sites. The map in Figure 4 clearly revealed that position P1 was near the edge of the road while positions P2–P5 were all on a path between two plantations, so different ground stiffness may be the cause of the occurrence of the abnormality in Figure 16a. An exponential attenuation process can be found in Figure 16a when fitting the tendency only through four sites located at the same ground condition. Figure 16b shows that the median-based distribution of the 13 monitoring zones at multiple intrusion sites was biased towards the positive direction of the *X’* axis, which substantially conformed to the response asymmetry observed in Figure 11, Figure 12, Figure 13, Figure 14 and Figure 15, namely, the offset angle between the actual intrusion path shown in Figure 4 and the defined *Y* axis in Figure 10 caused the asymmetrical distribution shown in Figure 16b. Compared with the threshold set to two, as depicted in Figure 16b, the median of 12 among 13 monitoring zones was not less than 2.08, although the perturbation responses from the small excavator were tiny during the entire experiment.

## 5. Conclusions

This study verified that ultra-weak FBG array is a viable method to meet the requirements of distributed dynamic measurement technology in actual engineering. The capability of detecting the ground intrusion in an underground structure based on such measurement technology was reported through field tests. The analysis indicated that the proposed approach has the potential to distinguish the strength and pattern of the typical load in the intrusion process within a certain range. Besides the detectability of the intrusion event based on variations in vibration amplitude, the location of ground intrusion can be inferred by the temporal and spatial distribution characteristics of vibration responses along the tunnel. In view of the approved test time, interferences of subway trains and ground transportation were not considered in the analysis, which seems to be a shortcoming of the study that deserves further attention. However, due to the fact that the load influence generated by the actual intrusion event is often greater than that of the small excavator adopted in this paper, it is believed that the proposed approach can be applied to identify the ground intrusion occurring in the daytime.

## Figures and Tables

**Figure 1 sensors-19-02160-f001:**
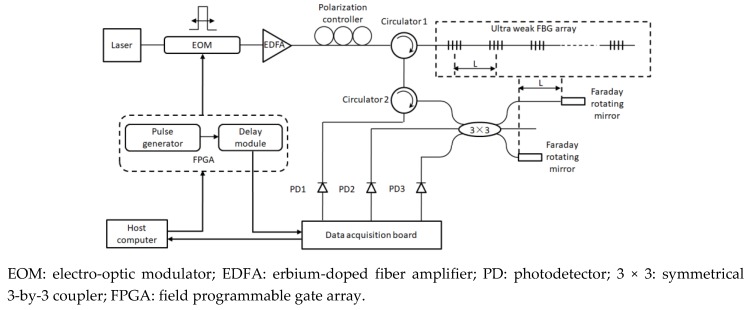
Sensing principle of distributed vibration detection based on ultra-weak FBG array.

**Figure 2 sensors-19-02160-f002:**
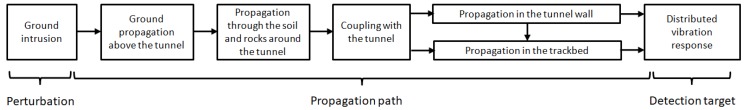
Propagation path of the intrusion load assumed by the detection principle.

**Figure 3 sensors-19-02160-f003:**
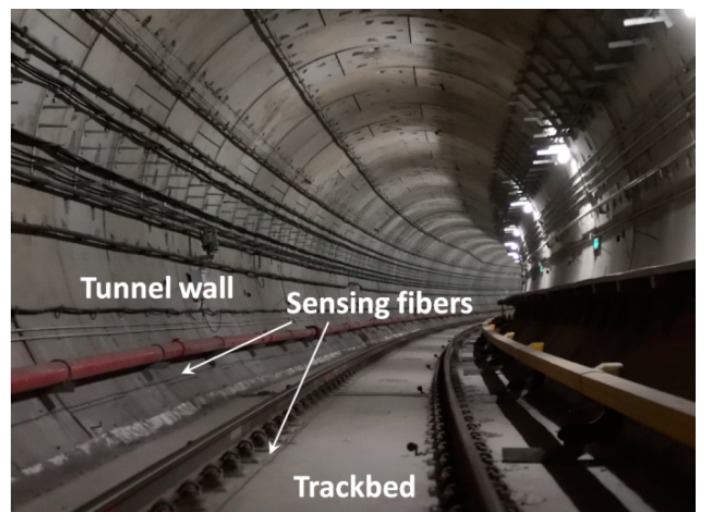
Distributed vibration sensing fibers mounted on the surfaces of the tunnel wall and the track bed.

**Figure 4 sensors-19-02160-f004:**
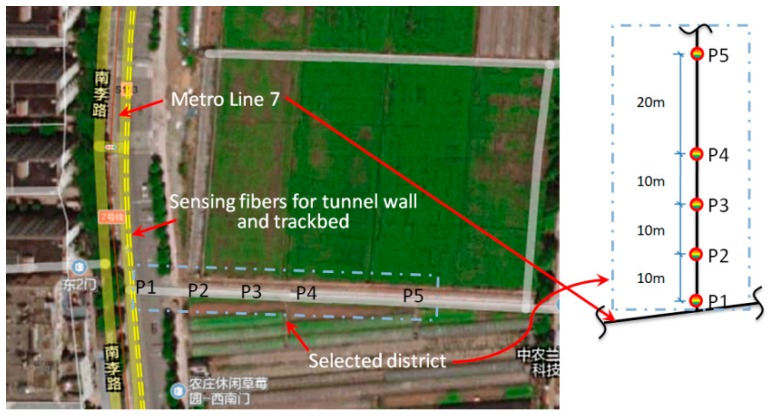
Sites and path of the simulated ground intrusion.

**Figure 5 sensors-19-02160-f005:**
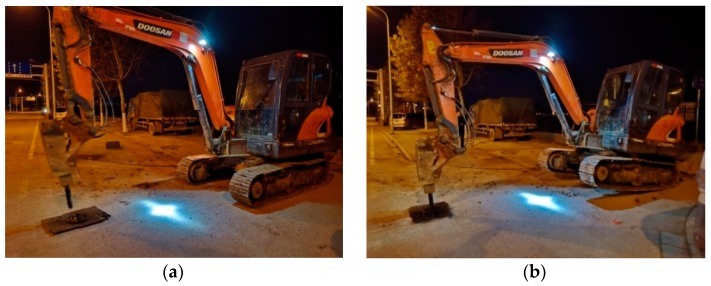
Two perturbation postures of the excavator: (**a**) fully and (**b**) partially touching the ground.

**Figure 6 sensors-19-02160-f006:**
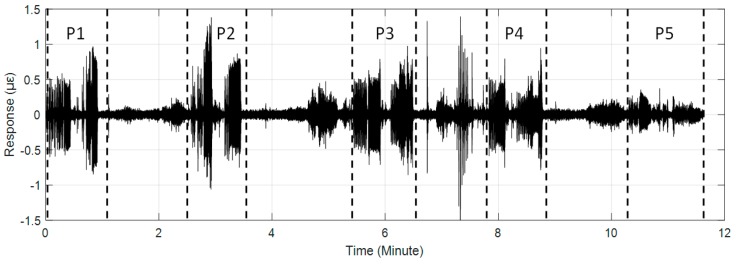
Distributed structural vibration of #159 zone of the tunnel wall under multiple position perturbations.

**Figure 7 sensors-19-02160-f007:**
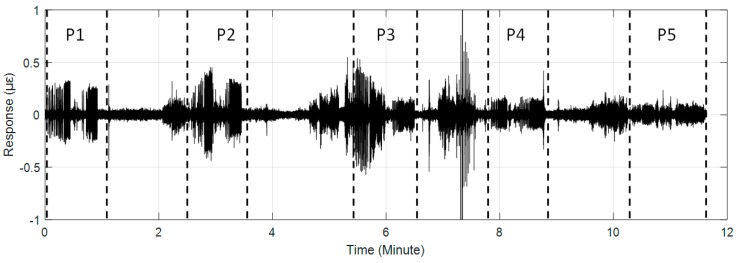
Distributed structural vibration of #159 zone of the track bed under multiple position perturbations.

**Figure 8 sensors-19-02160-f008:**
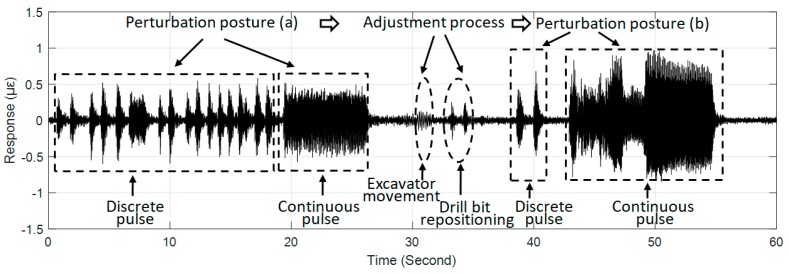
Distributed vibration response of #159 zone of the tunnel wall under the perturbation of position P1.

**Figure 9 sensors-19-02160-f009:**
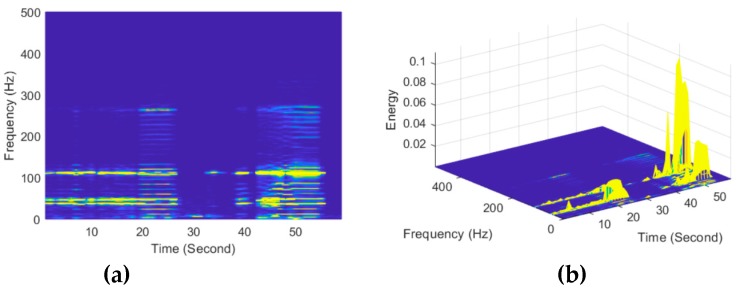
Time–frequency spectrum of #159 zone of the tunnel wall under the perturbation of position P1 viewed in (**a**) 2D and (**b**) 3D.

**Figure 10 sensors-19-02160-f010:**
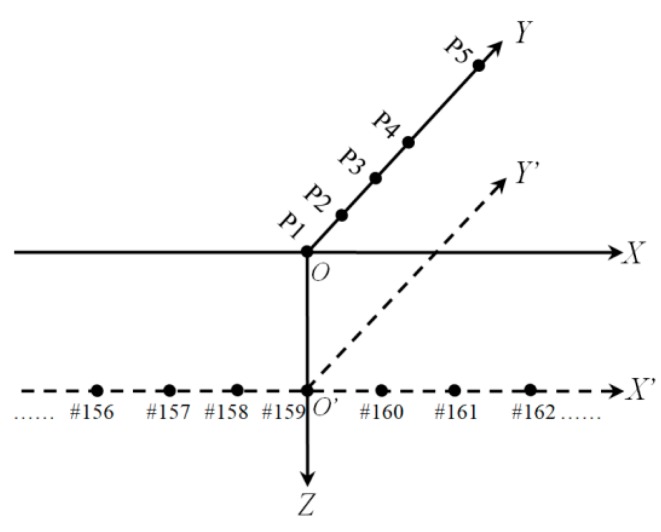
Coordinate system defined for the analysis of influence range of the simulated intrusion.

**Figure 11 sensors-19-02160-f011:**
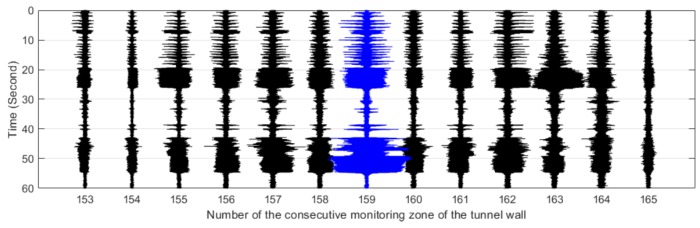
Distributed vibration responses of 13 consecutive monitoring zones under the perturbation of position P1.

**Figure 12 sensors-19-02160-f012:**
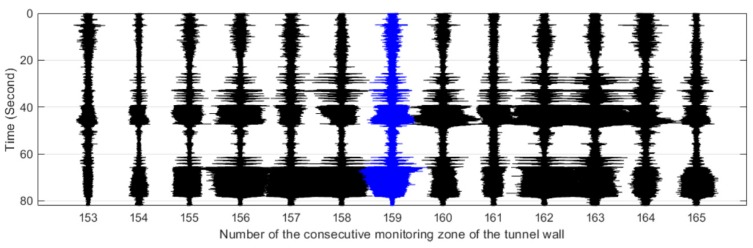
Distributed vibration responses of 13 consecutive monitoring zones under the perturbation of position P2.

**Figure 13 sensors-19-02160-f013:**
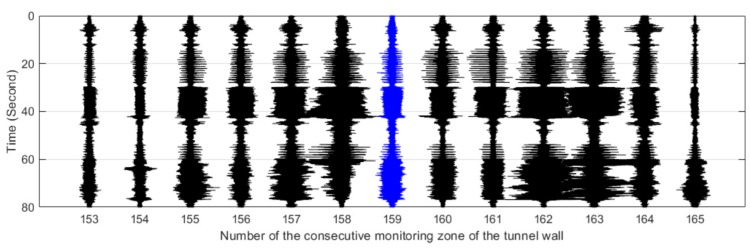
Distributed vibration responses of 13 consecutive monitoring zones under the perturbation of position P3.

**Figure 14 sensors-19-02160-f014:**
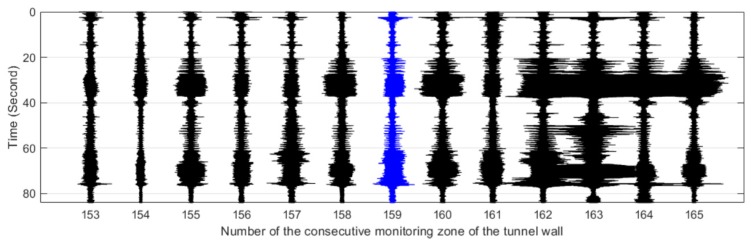
Distributed vibration responses of 13 consecutive monitoring zones under the perturbation of position P4.

**Figure 15 sensors-19-02160-f015:**
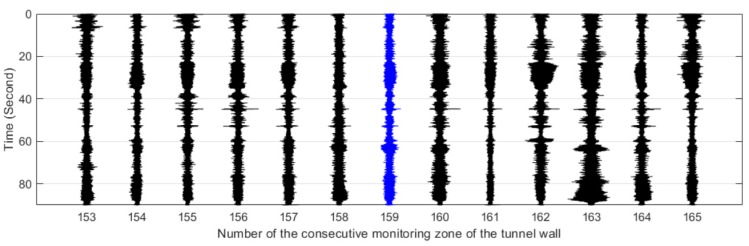
Distributed vibration responses of 13 consecutive monitoring zones under the perturbation of position P5.

**Figure 16 sensors-19-02160-f016:**
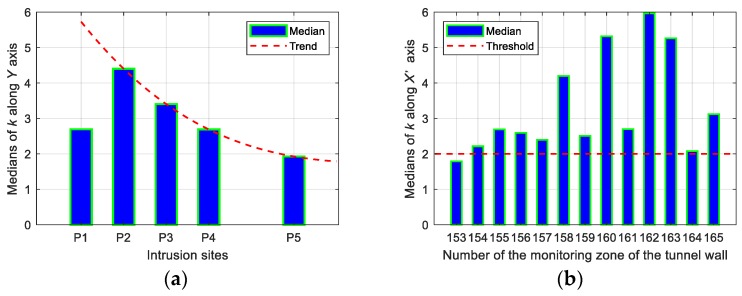
Median distributions along (**a**) the intrusion site and (**b**) the monitoring zone.

**Table 1 sensors-19-02160-t001:** Comparisons between common sensors for underground structure monitoring and ultra-weak fiber Bragg grating (FBG).

Sensors	Static/Dynamic Measurement	Multiplexing Capacity	Reflectivity index ^1^	Transmission Medium
Electronic	Both	Weak	N.A. ^2^	Electric cable
FBG	Both	Median	0.1–1	Sensing optic fiber
Rayleigh-based OTDR/OFDR	Static	Strong	10^−9^–10^−7^	Ordinary optic fiber
Brillouin-based BOTDR/BOTDA	Static			
Rayleigh-based DAS	Both			
Ultra-weak FBG	Both		10^−5^–10^−4^	Sensing optic fiber

^1^ Higher reflectivity index usually means better signal-to-noise ratio (SNR); ^2^ Not applicable.

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
