# Peer review of "Identification of Ground Intrusion in Underground Structures Based on Distributed Structural Vibration Detected by Ultra-Weak FBG Sensing Technology"

_sensors, 2019, doi:10.3390/s19092160_

Reviewer 1 Report

As it is presented, I need to ask for major revisions for this article and after resubmit it since there are a lot of papers including review papers where it was not mentioned and state of the art is not well addressed like:

 https://doi.org/10.3390/s17112511

https://www.sciencedirect.com/science/article/pii/S2095268617306122

Also, a final table to compare all technology in the literature must be added and see what are the main advantages compared with the already published. Also, ultraweak FBG technology is not new and there is no novelty in this paper and it needs to be addressed and well justified why it is good for publication in MDPI Sensors.

Author Response

Response to Reviewer 1 Comments

Point 1: As it is presented, I need to ask for major revisions for this article and after resubmit it since there are a lot of papers including review papers where it was not mentioned and state of the art is not well addressed like:

https://doi.org/10.3390/s17112511

https://www.sciencedirect.com/science/article/pii/S2095268617306122

Response 1:

Thanks to the reviewer's suggestions for further stating state of the art. In addition to the provided two papers, the revised manuscript added the following recent and other necessary literatures and reorganized the introduction part.

n  Jo, B.W.; Khan, R.M.A.; Lee, Y.S.; Jo, J.H.; Saleem, N. A fiber Bragg grating-based condition monitoring and early damage detection system for the structural safety of underground coal mines using the Internet of things. J. Sensors 2018, 2018, 9301873.

n  Hong, C.; Zhang, Y.; Lu, Z.; Yin, Z.A. FBG tilt sensor fabricated using 3D printing technique for monitoring ground movement. IEEE Sensors J. 2019, 19, 1590.

n  Pamukcu, S.; Cheng, L.; Pervizpour, M. Chapter 1 – Introduction and overview of underground sensing for sustainable response. In Underground Sensing. Monitoring and Hazard Detection for Environment and Infrastructure, 1st ed.; Pamukcu, S., Cheng, L.; Academic Press, 2018; pp. 1–42.

n  Lei, X.; Xue, Z.; Hashimoto, T. Fiber optic sensing for geomechanical monitoring: (2)- distributed strain measurements at a pumping test and geomechanical modeling of deformation of reservoir rocks. Appl. Sci. 2019, 9,417.

n  Miah, K.; Potter, D.K. A review of hybrid fiber-optic distributed simultaneous vibration and temperature sensing technology and its geophysical applications. Sensors 2017, 17, 2511.

n  Yang, M.; Li, C.; Mei, Z.; Tang, J.; Guo, H.; Jiang, D. Thousand of fiber grating sensor array based on draw tower: A new platform for fiber-optic sensing. In Proceedings of Optical Fiber Sensors, OFS 2018, Lausanne, Switzerland, 24-28 September 2018.

n  Bartelt, H.; Schuster, K.; Unger, S.; Chojetzki, C.; Rothhardt, M.; Latka, I. Single-pulse fiber Bragg gratings and specific coatings for use at elevated temperatures. Appl. Opt. 2007, 46, 3417-3424.

n  Ecke, W.; Schmitt, M.W.; Shieh, Y.; Lindner, E.; Willsch, R. Continuous pressure and temperature monitoring in fast rotating paper machine rolls using optical FBG sensor technology. In Proceedings of 22nd International Conference on Optical Fiber Sensors, Beijing, China, 15-19 October 2012.

n  Bai, W.; Yang, M.; Hu, C.; Dai, J.; Zhong, X.; Huang, S.; Wang, G. Ultra-weak fiber Bragg grating sensing network coated with sensitive material for multi-parameter measurements. Sensors 2017, 17, 1509.

n  Guo, H.; Qian, L.; Zhou, C.; Zheng, Z.; Yuan, Y.; Xu, R.; Jiang, D. Crosstalk and ghost gratings in a large-scale weak fiber Bragg grating array. J Lightwave Technol 2017, 35, 2032-2036.

n  Guo, H.; Liu, F.; Yuan, Y.; Yu, H.; Yang, M. Ultra -weak FBG and its refractive index distribution in the drawing optical fiber. Opt. Express 2015, 23, 4829-4838.

n  Gong, H., Kizil, M.S., Chen, Z., Amanzadeh, M., Yang, B., Aminossadati, S.M. Advances in fibre optic based geotechnical monitoring systems for underground excavations. Int. J. Min. Sci. Technol. 2017, 29, 229-238.

n  Tong, Y.; Li, Z.; Wang, J.; Wang, H.; Yu, H. High-speed Mach-Zehnder-OTDR distributed optical fiber vibration sensor using medium-coherence laser. Photonic Sens. 2018, 8, 203-212.

n  Luo, Z.; Wen, H.; Guo, H.; Yang, M. A time- and wavelength-division multiplexing sensor network with ultra-weak fiber Bragg gratings. Opt. Express 2013, 21, 22799-22807.

Point 2: Also, a final table to compare all technology in the literature must be added and see what are the main advantages compared with the already published. Also, ultraweak FBG technology is not new and there is no novelty in this paper and it needs to be addressed and well justified why it is good for publication in MDPI Sensors.

Response 2:

The reviewer's suggestion for adding a technical comparison table is great. This is very helpful to clearly highlight the research features of this paper. Although the ultra-weak FBG technology has been reported in some papers, the study using the ultra-weak FBG to perform dynamic measurement is still less, especially in addressing the problem of actual engineering. More studies conducted in laboratory or field primarily focus on the static measurement (such as strain and temperature, etc) based on ultra-weak FBG. In addition, the same condition exists in geotechnical engineering monitoring. For example, as the Table 1 listed in the first paper provided by reviewer, the main indicator parameters of interest still confine to strain and temperature. The distributed vibration measurement based on ultra-weak FBG for large-scale and long distance range discussed in this paper is different from the current monitoring research that based on fiber optic sensors in geotechnical engineering. In addition, compared with the similar DAS technology, the characteristic of optic fiber determines that the ultra-weak FBG has a higher signal-to-noise ratio under the same conditions. Therefore, according to the reviewer's suggestion, a technical comparison table was added at the end of the introduction part, elaborating the advantages of ultra-weak FBG technology in dealing with large-scale dynamic distributed measurement requirements.

Reviewer 2 Report

 The authors demonstrate an Identification of ground intrusion in underground structures based on distributed structural vibration detected by ultra-weak FBG sensing technology. Some of the experimental results are interesting. However, there are some points should be emphasized and interpreted. Therefore, the following comments may help the authors to modify the manuscript:

1.    The author must be developing a theoretical analysis of the sensor in order to explain how such a high sensitivity is achieved.

2.    How about the repeatability of the sensor?

3.    During the characterization procedure the authors have taken in account the polarization effects?

Author Response

Response to Reviewer 2 Comments

Point 1: The author must be developing a theoretical analysis of the sensor in order to explain how such a high sensitivity is achieved.

Response 1:

First of all, I would like to thank the reviewer for your positive recognition of the topic of this manuscript. For the sensing system used for identification of ground intrusion in subway presented in this paper, the co-author (the third author of this paper) conducted relevant basic theoretical research (Reference [33]), and gave detailed experimental design and theoretical analysis process. The conclusion part of the previous study shows that the system has the characteristics of high speed, high sensitivity and low fault tolerance, etc. Therefore, in order to ensure the integrity of the theoretical basis of the submit paper, and to avoid repeated descriptions, the revised manuscript made an annotation and citation in the second paragraph of Section 2.

Point 2: How about the repeatability of the sensor?

Response 2:

We made a citation (Reference [34]) and answered the repeatability indicator in the second paragraph of Section 2. The cited paper is also the recent experimental results of our group. Your question is full of constructiveness, which makes the theoretical support of the revised manuscript more complete.

Point 3: During the characterization procedure the authors have taken in account the polarization effects?

Response 3:

As we all known, polarization effect is an unavoidable problem in all research based on the principle of light interference. In this paper, in order to interrogate the signal of distributed vibrations, Faraday rotating mirrors were considered in the demodulation process of ultra-weak FBG array to weaken the polarization effect (see in Figure 1). We clarified this in the first paragraph of Section 2 in the revised manuscript. To be frank, whether there is a more appropriate method for suppressing the polarization effect is still under study. At current stage, the main goal of this paper was to report the feasibility of identifying ground intrusion based on distributed vibration of ultra-weak FBG, and the designed demodulation system basically met the need of identification of the simulated ground intrusion in subway. Our team will continue to conduct research on the elimination of polarization effects and strive to publish special research reports in the future.

Round  2

Reviewer 1 Report

The authors improved the paper with a good piece of work.